# A Socioecological Perspective of How Physical Activity and Sedentary Behaviour at Home Changed during the First Lockdown of COVID-19 Restrictions: The HomeSPACE Project

**DOI:** 10.3390/ijerph19095070

**Published:** 2022-04-21

**Authors:** Amie B. Richards, Masoumeh Minou, Michael P. Sheldrick, Nils Swindell, Lucy J. Griffiths, Joanne Hudson, Gareth Stratton

**Affiliations:** 1Applied Sports Technology Exercise and Medicine (A-STEM) Research Centre, Swansea University, Swansea SA1 8EN, UK; 657783@swansea.ac.uk (A.B.R.); m.minou@swansea.ac.uk (M.M.); michael.sheldrick@swansea.ac.uk (M.P.S.); n.j.swindell@swansea.ac.uk (N.S.); joanne.hudson@swansea.ac.uk (J.H.); 2Population Data Science, Medical School, Swansea University, Swansea SA2 8PP, UK; lucy.griffiths@swansea.ac.uk

**Keywords:** COVID-19, children, home, physical activity, sedentary behaviour

## Abstract

The COVID-19 pandemic forced school closures, resulting in home schooling, more time spent at home and fewer opportunities for physical activity (PA). This study explored factors influencing PA and sedentary behaviours (SB) within the home environment during the first lockdown, starting in March 2020. Twenty semi-structured interviews (20 parents and 23 children, 12 years ± 1.25) were conducted. Data were coded using thematic analysis on NVivo© and concepts from McLeroy’s socioecological model for health promotion were used to analyse the data. Findings indicate that children’s PA and SB at home were influenced by: (i) individual-level factors (e.g., gender, competence, attitudes and motivation); (ii) interpersonal-level factors (e.g., siblings, parents, pets, friends and coaches); (iii) organisation-level factors (e.g., school, clubs and societies), (iv) community-level factors (e.g., home and local environment, access to facilities, social norms, time constraints and home equipment), and (v) policy-level factors (e.g., lockdown restrictions). Stay-at-home mandates resulted in perceived reductions in PA and increases in SB within the home; however, this provided alternative positive opportunities for families, including more time to spend together and exploring green and blue spaces in the local area.

## 1. Introduction

Children spend most of their time at home and indoors [1], which is a concern as children are more sedentary when indoors [2]. In the year 2020, children were forced to spend a considerable amount of time in their home environment due to the COVID-19 global pandemic. Many countries employed a stay-at-home directive to reduce transmission of the virus and, as a result, many businesses moved to a working-from-home model, with the hospitality sector, non-essential shops, schools and other education settings being forced to close. School closures meant that children were home-schooled with remote, online learning provided by teachers, and this removed opportunities for PA including face-to-face physical education (PE), active transport to and from school, together with break and lunchtime activity.

According to the World Health Organisation (WHO), failing to regularly achieve PA recommendations and spending an increasing amount of time in SB is the fourth leading risk factor of mortality, only behind obesity, hypertension and tobacco use [3]. Children’s PA has been widely explored and the physiological and psychological benefits to children’s health have been well established [4,5], yet most children still choose sedentary activities as ways to spend their leisure time [6]. Moreover, it is estimated that only 17.5% of children in the UK meet the recommended level of 60 min of moderate to vigorous physical activity (MVPA) per day [7,8].

Recent research has explored children’s PA during the COVID-19 pandemic with worrying speculations that short-term decreases in PA levels and increases in SB may become permanent [9]. When schools were reopened, objectively measured PA levels had decreased and SB remained high [10]. Research with adults has also shown decreases in daily steps taken when a full lockdown was in place, whereas similar decreases in daily steps were not observed in areas where there was a partial lockdown [11]. These downward trends in PA are present, despite research identifying the importance of PA in reducing the risk of severe COVID-19 disease [12].

Qualitative studies have begun to explore the reasons for decreases in PA, finding that an increase in screen time [13] and the cancellation of organised activities [9] were two factors which negatively impacted children’s PA levels during COVID-19 lockdown restrictions. One study found that the main barrier to maintaining PA levels during the COVID-19 pandemic was not having access to an outdoor space or PA equipment [14]. Whilst outdoor spaces were of particular importance, during the COVID-19 pandemic, children’s home space was also a key factor as findings show that children’s PA at home either increased or stayed the same during the pandemic [15]; despite this, it did not compensate for the out of home reduction in PA.

Children’s PA at home was explored pre-pandemic with one study finding that almost 50% of children’s overall MVPA and sedentary time was accrued in the home [16], indicating that PA and SB are largely influenced by the home environment. Further research has found that an open-plan living area was positively associated with total PA and MVPA, whilst sitting breaks were positively associated with garden size, suggesting that those with bigger gardens had more opportunities to break up SB [17]. Research has concluded that the physical home environment can provide both barriers and facilitators to children’s PA and SB [18]. Despite the importance of the physical home environment, it was also highlighted that the home space is socially impacted by the people living in it, hence it is a dynamic ecological setting [18].

In this dynamic home context, both PA and SB can be influenced by multiple factors at different levels. This provides an opportunity to use the socioecological model to enhance understanding of children’s PA and SB, acknowledging that interactions between people and their environment are key factors [19]. This model proposes five levels: individual characteristics (e.g., sex, age, and motivations), interpersonal factors (e.g., family, friends), organisational factors (e.g., schools), community factors (e.g., social norms) and policy-level factors (e.g., policy and law).

The primary aim of this study was to improve understanding of the impact of COVID-19 lockdowns on children’s PA and SB at home using the socioecological model as a theoretical framework. A secondary aim was to make recommendations to improve children’s PA at home and their subsequent health, in the event of spending prolonged periods of time in the home environment. As such, qualitative research was carried out to identify barriers and facilitators of children’s PA and SB at home during the first lockdown of the COVID-19 pandemic, in the UK, between March 2020 and June 2020.

## 2. Materials and Methods

### 2.1. Study Design and Participants

The HomeSPACE study was a cross-sectional observational study investigating the influence of the home environment on children’s PA levels and sedentary time [17]. The HomeSPACE COVID-19 project is a longitudinal study investigating how homes have changed during COVID-19 and the impact on children’s PA and SB. During the second phase of the HomeSPACE COVID-19 project, families were invited to take part in a semi-structured, online video interview to explore PA and SB changes within the home resulting from COVID-19 lockdown restrictions between March 2020 and June 2020.

Participants were recruited to the qualitative element of the HomeSPACE study by categorising all 103 families into tertiles generated from the Welsh Index of Multiple Deprivation (WIMD). Participants were split into socioeconomic status (SES) tertiles based on the WIMD as follows: low- (1–636), medium- (637–1272) and high-SES (1273–1909) groups. Within each SES group, variables measured in the quantitative element of the study (MVPA, sitting time, house size and garden size) were stratified into three groups: high; medium, and low. The “stratified” function in R© (4.0, R Core Team, Vienna, Austria) was then used to allow for random sampling to select an equal number of participants from each stratum. These participants were subsequently contacted via email and telephone. Twenty families agreed to participate including 20 parents (90% female) and 23 children (39% girls) (aged 12 years ± 1.25). Thirteen families were from high, three from medium and four from low SES; this split was uneven due to the final sample of volunteer participants.

### 2.2. Data Collection

Following institutional Research Ethics Committee approval (REC: MS_2020-029a), participants were contacted to organise online interviews with families to include one parent and at least one child. Interviews were organised at a time suitable for the family, between June 2020 and August 2020, and were all conducted by the same researcher who was trained in qualitative research methodology. The interviews were recorded via Zoom© with the participant’s written consent. During the interviews, participants were asked to express their feelings and opinions on the effects of the COVID-19 restrictions on the children’s PA and SB within the home environment. A semi-structured interview guide was created and used to ensure that similar questioning routes were pursued with each participant, with flexibility to respond to lines of discussion raised by each participant. Question topics included most frequent activities, social elements including siblings, space within the home including the garden, routines and differences on weekend and weekdays. These questions were pilot tested with a convenience sample of similarly aged children to ensure understanding and fluidity prior to the interviews taking place.

### 2.3. Data Analysis

Once data collection was completed, interviews were transcribed using the automated transcriber on Zoom© (5.1.0, Zoom Video Communications, San Jose, CA, USA) and then checked and cleaned. Data were analysed using Braun and Clarke’s (2022) [20] thematic analysis process combining both inductive (data-driven) and deductive (actively searching for perceived factors affecting PA and SB at home) techniques. The process started with familiarisation, which involved reading and re-reading the transcripts and highlighting data of importance, including that which were repetitive across numerous interviews, related to previous research and suggested a novel finding. These significant data were then coded using NVivo 12© (NVivo12, QSR International, Melbourne, Australia). After reviewing the codes and grouping-related codes together, initial themes and sub-themes were generated in a hierarchical manner. Final theme names were conceived and substantiated by data obtained from the transcripts. Data were then deductively analysed in line with the socioecological model to map the themes in line with the model’s five levels: individual, interpersonal, organisational, environmental and policy. To ensure credibility, Lincoln and Guba’s (1985) criteria [21] were considered, keeping in mind Gergen’s (2014) evaluation that these are only useful under certain conditions and should be study specific [22]. The researchers engaged in prolonged engagement to immerse themselves in the research to understand multiple factors that were being investigated. Discussions with the research team, termed peer debriefing [21], together with self-reflection and progressive subjectivity were key in challenging thoughts and reviewing results. 

## 3. Results

### 3.1. Socioecological Model

The results were organised following the socioecological model, which showed the multiple factors involved in children’s PA and SB at home during the COVID-19 pandemic. There were factors identified that could be both barriers and facilitators at each level of the socioecological model: individual, interpersonal, organisational, community and public policy.

#### 3.1.1. Individual Level

##### Gender

Boys and girls showed differences in their PA and SB at home during the lockdown, with girls choosing more sedentary activities, as highlighted by one parent.

“I’m a teacher in secondary school and I find it really annoying and even seeing my son like they’ll go to the beach now and they’ll all take a ball…and then make up games…whereas the girls don’t.”(Mother, High SES)

##### Competence

Children’s physical competence in relation to PA was also a factor that influenced their PA levels at home during the lockdown, with children who were more physically competent remaining more physically activity during the lockdown.

“Yeah, I suppose, sport is his passion as much as anything. And that’s what he does excel at in he’s very good and he gets a lot of enjoyment…so that’s kind of a bit motivator for him.”(Mother, High SES)

##### Attitudes, Motivation and Enjoyment

Although many children reported that enjoying PA made them want to be physically active, others stated that they were motivated to be physically active for several different reasons, including keeping fit and healthy, mental health and wellbeing benefits and because their family were physically active. One child spoke about setting goals to maintain their motivation to be physically active during lockdown. These positive attitudes and motivations towards PA facilitated greater levels of PA than those children who reported negative attitudes or a lack of enjoyment of PA.

“Wanting to like exercise and stuff it makes me happier in general.”(Girl, Aged 12, High SES)

“Well, it’s mostly just things I like to do so. I quite like to run around in the park and play with my dad and brother. And then if it’s anything to do with like I like going to like public pools and splashing about there. I like going to the beach as well just jumping in the sea.”(Boy, Aged 12, High SES)

“I just love doing sports and it keeps me active all the time.”(Boy, Aged 12, Low SES)

“Well, I’ve set myself a goal in running and so that that keeps me active. Yeah, I like setting myself goals so that helps to motivate.”(Boy, Aged 15, Medium SES)

#### 3.1.2. Interpersonal Level

##### Siblings

Children discussed that they spent more time with their siblings, and this helped them to be more physically active during the COVID-19 lockdown restrictions.

“With brother and sister like they always find to do things or just even like jumping with each other.”(Mother, Low SES)

“Yeah, I would probably mess around the house with my brother for a bit and play some sports in the back garden sometimes.”(Boy, Aged 13, Medium SES)

“Yeah, we played a lot more [with my siblings] because one of them was in uni [university] but has now moved back so we played with him more.”(Boy, Aged 13, High SES)

##### Parental Support

Parental support was a key theme that was generated in this study. Having parents who were supportive of PA promotion was one factor that helped in preventing children from spending too much time pursuing SB.

“So, she had to go and measure how far she can run or something. So my husband and I went to the park to do that.”(Mother, Low SES)

“So anytime we could get him off the games and out into the garden or at least you know outside we would do.”(Mother, High SES)

“And sometimes I’d complain to my mom that I was bored, and she would tell me to go for a run.”(Boy, Aged 13, Low SES)

“Our mom and dad tried to get us out all time.”(Boy, Aged 13, High SES)

##### Pets

Families with dogs had more motivation to be physically active, illustrating the link between family environment and intrapersonal factors. Both adults and children had a reason to get out of the house and take their dog for a walk, therefore decreasing their sedentary time. 

“Well, my neighbours got a new puppy, so I’d go out and play a lot.”(Girl, Aged 12, High SES)

“Yeah, exactly and he did play an awful lot with the dog, poor dog is exhausted you know he was in the garden. He’s had enough now it’s all that kind of thing that they do constantly. I’d say…the dog’s been a big source of company and exercise for him as well.”(Mother, High SES)

“I’ve got a Yorkshire Terrier and she loves him, completely adores him so it’d be something that…you know she’ll take him you know for walks. That was something that motivates her.”(Mother, Low SES)

“And so, we were using more leaving the house as sort of the reason for physical activity. We…had the dog that we had to walk him. So that was a good excuse to get out and get some exercise.”(Mother, High SES)

##### Friends and Peer Influence

During the lockdown, friends and peer influence seemingly had a greater impact on SB, due to the opportunities to play with friends remotely that online gaming provides. Due to restrictions, children were prevented from socialising in person, so many turned to online gaming to stay connected with their peers. However, this increased their screen time and therefore their SB.

“Yeah, like, mainly because I think one of the reasons was because I couldn’t see my friends in the day I play Xbox with them.”(Boy, Aged 13, High SES)

“I felt like I was missing out because I wasn’t playing [online] with them. I felt like I was missing out and stuff so I got more like I wanted to stay home.”(Boy, Aged 13, High SES)

“I think he spends more time on screen because his friends are on screen as well.”(Father, High SES)

##### Community Coaches

An additional interpersonal factor is the influence of community coaches on the children’s PA and SB. If the child attended a community sports club, some coaches tried to continue with their coaching virtually during lockdown. This increased the children’s opportunities to be physically active.

“Well, our coach had the idea and whoever did the most of one type of skill would get a prize or something. And things like different types of tricks. And so, we had like a week or so to do it. And then we put in our scores and then whoever did a certain amount would be put in their name would be put in a hat. And then he would pick out a random name they would get the prize.”(Boy, Aged 12, High SES)

##### Family Routine

A primary theme from the interviews was family routine and how this changed through the pandemic restrictions. Those families who maintained behaviours seemed to remain more physically active than those whose behaviours changed as a result of the restrictions.

“No, my mom woke me at the same time I would normally.”(Girl, Aged 12, High SES)

“Yes, more structure to the day, like times mattered [pre-pandemic] whereas in lockdown it was just like another day.”(Mother, High SES)

“Once we got into a routine, he did it without too much moaning.”(Father, High SES)

“At the start of lockdown, we tried to get them into some kind of routine.”(Mother, High SES)

#### 3.1.3. Organisational Level

##### Schools

A key organisational-level theme was the role of schools during lockdown. Schools had differing approaches to providing home schooling. Some schools provided live online home-schooling lessons and they monitored the children’s progress whilst others set work for the children to complete remotely and submit. Despite PE being a compulsory part of the curriculum, some schools did not provide PE lessons, whilst others provided opportunities for PA as a replacement, including YouTube videos for the children to workout to, or yoga. Overall, home schooling increased children’s screen time and SB as seen through the quotes below.

##### Home Schooling and Screen Time

“All of her education was done through the laptop and the school were pretty good actually maintaining a regular school day.”(Father, High SES)

“When I was doing my work, I would mostly be on the screen.”(Boy, Aged 13, High SES)

“And so, I suppose [screen time] has probably gone up but I think his generally screen use has just gone up anyway because he’s doing a lot of learning online.”(Mother, High SES)

“I think it’s [screen time] gone higher since the lockdown because it’s been a lot of online training with school and she’s been there since the time that schools open at eight o’clock…until about possibly three, four in the afternoon you know, she’s, they’ve been doing, sending a lot of homework. So, she’s been on the screen, more than before. And it’s because of this, the lockdown.”(Mother, Low SES)

##### Physical Education (PE)

Discussion of PE reinforced the differing approaches taken by schools to providing PE for the children during the home-schooling period. There appeared to be limited direction on being physically active from schools. For some students, it was not included in their timetable:

“They tend to have a timetable which is one kind of subject a day and it was like one was humanities, one was science, one was maths, English and then I think like arts and crafts type thing. So no, no physical activity included.”(Mother, High SES)

For other children, the PE work was theoretical but not practical.

“I’m not aware of him having, I think he had some kind of theoretical stuff to do. But I was never aware of any practical stuff.”(Mother, High SES)

“Yeah, she had P.E homework, but it wasn’t anything to do with physical activity and was just quizzes.”(Mother, Medium SES)

Some of the children were asked to carry out activities either using online videos or alternative equipment which would not normally be available at school.

“Yes, they did. XXXX had whenever she should have had PE there was an hours worth of things in there, yoga and some exercises to home. XXXX also had a similar thing he had things to do you know that list of different YouTube videos to watch that they work out to.”(Father, High SES)

“I think it was there was more theory involved, but there was also, I know there was a couple of things. She was told to do. And she said she didn’t have the equipment so she’d kind of substitute did it for the trampoline or cycling or something that she skateboarding. She was still doing some form of physical exercise.”(Mother, High SES)

##### Clubs and Societies

Being part of a club or society often provided opportunities for the children to be physically active. This is linked to the interpersonal factor of community coaches. It is not just the coaches that had an impact here, the clubs themselves gave children the opportunity to stay connected with their team and/or group of peers involved in these clubs/societies who also took part in the virtual challenges. 

“We had to walk to Scotland or something that the team had to go to Scotland…everybody had to do so many miles or kilometres and but of course XXXX only did what he was supposed to do he didn’t do very much more.”(Mother, Low SES)

#### 3.1.4. Community Level

##### Physical Environment

Parents and children commented on the effect of their physical environment. This included both the home environment and the local environment, particularly when the restriction was posed that the public in Wales could not travel more than 5 miles from their home unless in extenuating circumstances. Families commented on how, before the 5-mile restriction, they were able to drive to safe outdoor spaces including parks and beaches for the children to be physically active; but once the 5-mile rule was introduced, this became illegal. However, this restriction gave families the opportunity to explore their local areas more and find areas that they were unfamiliar with.

“So, we go on lots of family walks, I would go walking with them or XXXX would take him walking and when the five-mile restriction was in we couldn’t go anywhere. It was lots of walks from the house”(Mother, Medium SES)

“There was no traffic, we were able to walk along the road it was so lush, so so lovely. And she actually blossomed during that time, because no peer pressure, can do whatever she wanted it was really lovely. So, when you know in terms of making a difference. Yeah, we’d always make sure that we would do something nice on the weekend that we’d go for a big walk.”(Mother, High SES)

##### Access to Facilities

Access to facilities was another community theme. Parents and children commented on the lack of community facilities available due to the lockdown restrictions. The leisure facilities children would normally use to be physically active were unavailable and this decreased their opportunities to be physically active.

“The only problem was when it was raining, we couldn’t go to like places like the LC2 or Limitless which made us sit more.”(Boy, Aged 12, High SES)

##### Weather

The first lockdown started in March 2020 as the spring and summer months were approaching and it was made clear that the children were more physically active during the warmer and drier weather.

“Definitely the weather, I won’t go out if it’s like raining or anything. I, even if it’s sometimes too cold. I don’t wanna go out because I can’t deal with it.”(Girl, Aged 12, High SES)

“If it’s sunny outside then we are normally always outside.”(Girl, Aged 12, Medium SES)

##### Outside Space

Together with the home environment and access to PA equipment within the home, the outside space at home influenced home-based PA during the lockdown. The size and scope of garden space within the individual’s home played a part, suggesting that those with a bigger or flatter garden spent more time outside and being physically active in the garden than those with smaller or less accessible garden spaces.

“And there’s not really space outside in my garden to do any like sports, because it’s quite a small garden.”(Girl, Aged 13, Low SES)

“Well, I’ve definitely been going for runs around where I live and maybe going on more walks as a family. And playing in the back garden more than I was before.”(Girl, Aged 13, High SES)

##### Social Norms

Another theme within the community level were social norms, within which there are two factors that were identified: time constraints and home-based equipment. There was a social norm around the time that children should allocate to learning, sleeping and leisure time, with learning, when at school, taking up most of a child’s day. Schools being closed meant that children had more freedom with their day, albeit they should have been taking part in home schooling; however, only a small proportion of a child’s day was spent home schooling.

“Yeah [I’ve had more time]. I wouldn’t really be messing around the house with my brother as much as I wouldn’t have time like that.”(Boy, Aged 13, Medium SES)

“Yeah, the actual time together as a family. Is a lot more limited. So the older ones have been moaning that we’ve done more in terms of going out for walks than we did before, just because we’ve got time in an evening, whereas normally you wouldn’t have it.”(Mother, High SES)

“And during the week [pre-lockdown] he would never have the computer on he would never have time to go on the computer.”(Mother, High SES)

“We also go on the trampoline a lot more because I wasn’t doing work all day.”(Boy, Aged 13, High SES)

##### Availability of and Access to Equipment

Many children highlighted that having PA equipment accessible at home helped them with being physically active during the lockdown. Media equipment accessibility was also noted as a facilitator to SB. Children and parents commented that having electronic equipment such as game consoles, phones and laptops increased the children’s SB. On the other hand, having bikes, basketball hoops, swings and scooters increased the children’s PA. Many families had to buy new electronic equipment during lockdown to fulfil the requirements of home schooling, increasing interest in screens and SB, potentially leading to a decrease in PA. Ultimately, parents had the financial power of what to buy, making a clear interaction with the interpersonal level of the model.

##### PA Equipment

“Paddling pool was up, she’d go out on her own. And play on that and jump on the trampoline.”(Mother, High SES)

“Yeah, and he has a trampoline, so he went on the trampoline. And also, he has a push up bar and pull up bar and he also likes a little bit of ball play as well.”(Mother, Low SES)

“And in the lounge where I know it’s a random place to have it but we got them out at the start of COVID, they were pushed to one side in the conservatory till then there’s a sit up hubs crunchy thing and a stepper of sorts. And again, you walk in and quite happily finding watching TV whilst on the stepper.”(Father, High SES)

“Only really that we had more of the exercise equipment out.”(Father, High SES)

##### Media Equipment

“XXXX had a laptop bought for because we didn’t have enough equipment to use.”(Mother, High SES)

“New TV we bought when we were in lockdown because they both wanted to watch their own TV programmes and there was a bit… fighting so at the beginning of lockdown we bought a telly.”(Mother, Low SES)

“I mean we signed up to things like Netflix.”(Mother, High SES)

“We had to buy a new laptop because XXXX was at a stage where she was using my laptop.”(Mother, High SES)

#### 3.1.5. Policy Level

The overarching theme was lockdown restrictions at a policy and legal level. The restrictions in place to curb the spread of COVID-19 meant that many children were unable to achieve their usual levels of PA due to missing activities such as active transport to school, PE lessons, attending sports clubs and non-organised play or PA with their friends.

“Before COVID restrictions. I was doing clubs in I went to Ju Jit Su and I was looking for an acting club to do but I can’t do them now because it’s very full contact.”(Girl, Aged 11, High SES)

“She does do Guides so she’s involved in Girl Guiding…but obviously that switch to Zoom at the moment. So again, it’s not as physically active as it normally would be.”(Mother, High SES)

## 4. Discussion

The aim of this study was to improve understanding of the impact of COVID-19 restrictions on children’s PA and SB at home, using the socioecological model as a theoretical framework. The results show that individual-level factors (gender, competence, attitudes and motivation), interpersonal-level factors (siblings, parents, pets, friends and coaches), organisational-level factors (school, clubs and societies), community-level factors (home and local environment, access to facilities, social norms, time constraints and home equipment) and policy-level factors (lockdown restrictions) influenced children’s PA and SB at home during the first lockdown of the COVID-19 pandemic. This model can be used to promote PA within the home by focusing on the facilitators explored within the results.

The views of the parents and children suggest that there was a decrease in overall PA and an increase in SB during the lockdown restrictions, which has previously been reported within a Canadian population [15]. McCormack et al. (2020) found that over three-quarters (75.9%) of children increased their general use of screen-based devices and that over half (52.7%) decreased time playing at the park; however, PA at home either increased (48.8%) or remained unchanged (32.9%) [15]. This suggests that PA at home increased due to lack of opportunities to partake in PA elsewhere. Despite this, one study from Ireland found no changes in adolescent girls’ reported PA during the lockdown restrictions [23]. Studies using objective measures in multiple countries and age groups, including 4–6 year olds in Spain [24], 7–12 year olds in Holland [10] and in the USA [25] all found decreases in PA and increases in SB throughout the lockdown restrictions. One reason for these declines in PA, particularly MVPA, included pandemic-related social isolation [26]. This was also true for this study, as many children reported being isolated meant that they spent more time interacting virtually with their friends in sedentary pursuits including online gaming, and less time engaging in PA as a result. The restrictions meant that the social elements that encouraged many of the children to partake in PA such as being part of a team, were no longer a part of being physically active.

Consistent with previous work [18], the findings indicate that the home is a dynamic environment and that multiple factors at all levels of the socioecological model influenced family PA and SB during the first lockdown of the COVID-19 pandemic. The home physical environment can present barriers to, and facilitators of, PA and SB; however, as previously suggested, it is the family living within the home that changes the dynamic of the environment. This was particularly evident within this study as families were brought together through working from home, returning from universities and home schooling. Changes in the condition of who was living at home and spending more time at home were both a barrier and facilitator to PA. Previous research found that having a sibling who participates in PA is positively associated with higher levels of PA than being an only child [27]. In this study, siblings were spending more time together seemingly increasing their PA through unstructured play, which also has many benefits for a child’s physical, emotional and social wellbeing [28]. A systematic review [29] concluded that although researchers suggest that family is the most important aspect to consider when exploring PA behaviours and attitudes [30], much of the research has been investigative of parental behaviours as opposed to siblings. Other researchers have explored the sex composition of parent–child and sibling dyads and also the birth order of siblings, with mixed findings including girls with brothers participating in more PA than girls with sisters [31]. Older siblings have been identified as role models for younger siblings, meaning that if the older sibling is physically active, the younger sibling will follow suit [32]. Together with the influence of siblings on PA and SB, parents also have an important role to play. This study found that those children with parents who were supportive of PA and provided greater PA opportunities engaged in more PA. In support of this, one COVID-19-based study found that children of parents who were more anxious about COVID-19 visited the park less often and were more likely to spend ≥2 h/day taking part in screen-based activities such as gaming, compared with children of less anxious parents [15].

The results highlighted the importance of the availability and accessibility of space and equipment. These are two key factors that have been previously reported to promote or inhibit PA during the COVID-19 restrictions [14]. Spending time outdoors can have multiple health benefits [33]; however, it was noted that during the lockdown, some families had limited access to outdoor spaces such as gardens whilst others enjoyed exploring their local areas within the 5-mile boundary of the restrictions. In many areas playgrounds were closed, whilst the closure of sports centres and community halls further decreased the availability and accessibility of PA promoting facilities. The accessibility and availability of PA-promoting, or indeed SB-promoting equipment, were widely discussed with both children and parents talking about new equipment that was bought to meet the needs of their new lifestyle. Many families bought new media equipment or subscriptions including laptops, televisions, gaming consoles and streaming services, inevitably increasing the opportunities for SB. Physical activity equipment was also purchased including bikes, roller skates and weights in an attempt to increase PA opportunities at home. Although our data cannot conclude the purchasing of new equipment has widened inequalities as it was not explored in a systematic way and the sample of SES was skewed due to volunteer convenience, future research could explore this in more detail. Together with the availability of this equipment, its accessibility was also a factor that was discussed. Within the physical home space, families moved equipment out of cupboards or unused rooms to make it visible and therefore more accessible, thus increasing PA at home.

Despite the limited structure to the children’s days, those families that reported keeping a structure and routine to the day seemingly maintained a higher level of PA. Parents reported that every day felt the same and it was difficult to discern weekdays from weekends and school holidays. This presents challenges for maintaining children’s PA and decreasing their SB as previous research has found that on average children engage in less PA and more SB on weekends compared to school days [34]. Similarly, research has found that children who are not enrolled in holiday camps or activities gain more weight over the school holidays than those who are enrolled [35]. A similar scenario was apparent during lockdown, where there was a lack a structure and routine; this could have potentially posed similar health risks, although this was not measured during this study.

All of the barriers and facilitators to PA and SB at home that have been previously discussed are complex and cannot be understood or addressed if levels of the model are studied in isolation. Exploring the interactions and reciprocal causation between the levels allows for a deeper understanding and various levels to be targeted to intervene more effectively. A dynamic systems approach may be needed that accounts for the dynamic changes in the factors that lead to a greater understanding of how to support PA and SB management in children within their home environment. Figure 1 shows the multiple factors at each level and the interactions between these. For example, one interaction includes motivation, attitudes and enjoyment at the individual level; however, various factors at the interpersonal level, including siblings, pets and parents, all seemingly have an impact on a child’s motivation, attitudes and enjoyment of PA and SB. It is also important to take into consideration interactions within levels together with between levels. An example of an interaction within levels is that between the individual factors of gender and competence, as it has been previously reported that boys are more competent than girls in the majority of fundamental movement skills [36]. Considering and exploring interactions within the model allow for a more holistic interpretation of the results and subsequently more effective interventions.

When interpreting the findings from this study, several strengths and limitations should be considered. To the authors’ knowledge, this is the first qualitative analysis of children’s PA and SB at home during the first lockdown of the COVID-19 pandemic in the UK, where both parents and children shared their thoughts and experiences of children’s PA and SB during lockdown and exclusively in the HomeSPACE. The semi-structured interviews allowed for detailed discussions with participants to delve deeper into their behaviours and attitudes to PA and SB during the first lockdown. The nature of the sequential interviews also allowed the researchers to develop the questions as the interviews progressed. However, this study was not without its limitations, including the sample of participants recruited. Of the twenty parents interviewed, only two of these were fathers. Similarly, although sampling aimed to achieve an equal split of families across SES levels, due to the volunteer uptake, there were more families from high socioeconomic backgrounds; this overrepresentation of high-SES families is common in research due to the difficulty in recruiting lower-SES participants [37,38]. Despite the over-representation of high-SES families within this study, the findings may be generalisable to areas which share similar geographical and home environment characteristics. This study also only focused on PA and SB during the first lockdown, where factors such as good weather, novelty of a new living situation and uncertainty about the future need to be considered. Future research should seek to evaluate reasons for changes in PA and SB at home over the subsequent COVID-19 restrictions that were put in place including the easing of restrictions. A focus should be placed on inequalities in children’s PA and SB, seeking to recruit a larger sample from lower-SES families, to identify any disparities in behaviour. The potential long-term effects of not being in school and having a routine or structure to a day are yet to be explored; future research should focus on the changes that were made in the first lockdown and whether they were sustained over time.

## 5. Conclusions

An interaction of factors together with the dynamic nature of the home environment presented both barriers and facilitators to children’s PA and SB at home during the first lockdown of the COVID-19 pandemic. Use of the socioecological model allowed us to clearly identify the level of influence of different perceived barriers and facilitators. Furthermore, the interaction between these levels provided an integrated view of the parent’s and child’s perceived changes to PA and SB at home during the lockdown. The policy-level factor of COVID-19 laws and restrictions provided the basis for this study as, without these regulations, children would have been continuing as normal. It was this factor that forced changes to be made to daily life and therefore PA and SB at home. Other levels of the model included multiple factors which seemingly influenced PA and SB at home during the first lockdown of the COVID-19 pandemic, including routine, availability and accessibility of suitable facilities, space and equipment, family members and the children’s attitudes towards PA and SB.

The results of this study provide key information to increase children’s PA at home. The findings can further inform interventions that seek to promote children’s PA at home. Moreover, home-based interventions should be developed in the event of future lockdowns. It is essential that the changes reported in this study do not become permanent and that children re-engage in PA opportunities when permitted.

## Figures and Tables

**Figure 1 ijerph-19-05070-f001:**
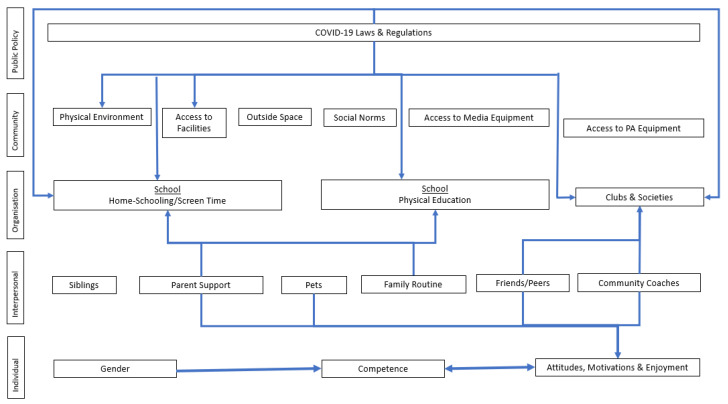
The multiple interactions between and within levels of the socioecological model exploring facilitators and barriers to children’s PA and SB with the home environment during the COVID-19 lockdown.

## Data Availability

The data are available to the research team according to ethical approval. The corresponding author is happy to provide data if required for scrutiny.

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
