# Peer review of "A Socioecological Perspective of How Physical Activity and Sedentary Behaviour at Home Changed during the First Lockdown of COVID-19 Restrictions: The HomeSPACE Project"

_ijerph, 2022, doi:10.3390/ijerph19095070_

Round 1
Reviewer 1 Report
This paper investigates the effects of the first lockdown on physical activity levels and sedentary behaviours of children, and provide recommendations to increase physical activity in order to improve general health.
It's a timely topic and the results are interesting. However, I have some suggestions to improve the paper.
In the introduction section, the authors should add a paragraph regarding the beneficial effects of physical exercise for health.
Moreover, several works reported the important role of physical activity also in COVID-19 affected patients。
Regarding the additional references suggested by me, please feel free to add only those that you deem appropriate in the context of your work. It is by no means mandatory to include these references or others for the revision to be reconsidered for publication. This is only a suggestion to improve the paper and to strengthen your statements to help better readers' understanding.
Author Response
Thank you for taking the time to review this manuscript and for your comments on the paper. We have considered these comments and have the following responses:
- With regards to the additional paragraph detailing benefits of physical exercise for health, we believe that in line number 42 we have identified some generic health benefits of PA, and PA is more of a focus in our study than exercise itself.
- In the introduction we have added a sentence to the paragraph about COVID-19 and PA: "These downward trends in PA are present, despite research identifying the importance of PA in reducing the risk of severe COVID-19 disease" (Sallis et al, 2021).
Reviewer 2 Report
Thank you for allowing me to review this paper entitled "A socioecological perspective of how physical 2 activity and sedentary behaviour at home 3 changed during the first lockdown of covid-19 4 restrictions: the homespace project". This study aims to explore how different factors have influenced physical activity and sedentary behaviours within the home environment during the first lockdown due to COVID19 pandemic.
The authors perform a cross-sectional observational study conducting twenty semi-structured interviews. The topic is interesting and the manuscript is well performed. The introduction section is clear. The methodology is well described and the results are shown clearly. The limitations are correctly listed in the discussion section. I have no additional remarks unless my sincere congratulations to authors.
Author Response
Thank you for taking the time to review this manuscript and for your comments on the paper. No action was needed to be taken.
Reviewer 3 Report
In the Introduction the model should be explained earlier in the explanation.As it is mentioned, this is a limited study that can not give a conclusion but can be used as a model to assess sedentary behaviors moving forward. The conclusions in this paper clearly only apply to a more affluent population, and therefore the conclusions can only apply to this one small cohort. Perhaps, the study should suggest that this model could be used to promote PA, and shift the perspective slightly to encompass a broader population than this small observed cohort.
Author Response
Thank you for taking the time to review this manuscript and for your comments on the paper. We have considered your comments and have the following responses:
- We considered moving the model explanation forward within the introduction. However, we believe that the order of the introduction is important in setting the overall scene for the paper.
- We have identified the skewed socioeconomic status (SES) of the participants as a limitation in the research. We have also included the SES of the individual participants within the results quotation section to ensure that readers can recognise the SES of the participant making the comment.
- L552 – We have added two references to justify the limitation of the over-representation of high SES families.
- L552 – We have added a sentence about the generalisability of the study – “Despite the over-representation of high SES families within the study, the findings may be generalisable to areas which share similar geographical and home environment characteristics.”
- We have eluded to the fact that the model could be used to promote PA, but have made this more explicit since your comment.
- L445 - "This model can be used to promote PA within the home by focusing on the facilitators explored within the results."